# REGRET BOUNDS AND REINFORCEMENT LEARNING EXPLORATION OF EXP-BASED ALGORITHMS

## ABSTRACT

EXP-based algorithms are often used for exploration in multi-armed bandit. We revisit the EXP3.P algorithm and establish both the lower and upper bounds of regret in the Gaussian multi-armed bandit setting, as well as a more general distribution option. The analyses do not require bounded rewards compared to classical regret assumptions. We also extend EXP4 from multi-armed bandit to reinforcement learning to incentivize exploration by multiple agents. The resulting algorithm has been tested on hard-to-explore games and it shows an improvement on exploration compared to state-of-the-art.

## 1 INTRODUCTION

Multi-armed bandit (MAB) is to maximize cumulative reward of a player throughout a bandit game by choosing different arms at each time step. It is also equivalent to minimizing the regret defined as the difference between the best rewards that can be achieved and the actual reward gained by the player. Formally, given time horizon $T$, in time step $t \le T$ the player choose one arm $a_t$ among $K$ arms, receives $r_{a_t}^t$ among rewards $r^t = (r_1^t, r_2^t, \ldots, r_K^t)$, and maximizes the total reward $\sum_{t=1}^T r_{a_t}^t$ or minimizes the regret. Computationally efficient and with abundant theoretical analyses are the EXP-type MAB algorithms. In EXP3.P, each arm has a trust coefficient (weight). The player samples each arm with probability being the sum of its normalized weights and a bias term, receives reward of the sampled arm and exponentially updates the weights based on the corresponding reward estimates. It achieves the regret of the order $O(\sqrt{T})$ in a high probability sense. In EXP4, there are any number of experts. Each has a sample rule over actions and a weight. The player samples according to the weighted average of experts' sample rules and updates the weights respectively.

Contextual bandit is a variant of MAB by adding context or state space $S$. At time step $t$, the player has context $s_t \in S$ with $s_{1:T} = (s_1, s_2, \ldots, s_T)$ being independent. Rewards $r^t$ follow $F(\mu(s_t))$ where $F$ is any distribution and $\mu(s_t)$ is the mean vector that depends on state $s_t$. Reinforcement Learning (RL) generalizes contextual bandit, where state and reward transitions follow a Markov Decision Process (MDP) represented by transition kernel $P(s_{t+1}, r^t | a_t, s_t)$. A key challenge in RL is the trade-off between exploration and exploitation. Exploration is to encourage the player to try new arms in MAB or new actions in RL to understand the game better. It helps to plan for the future, but with the sacrifice of potentially lowering the current reward. Exploitation aims to exploit currently known states and arms to maximize the current reward, but it potentially prevents the player to gain more information to increase local reward. To maximize the cumulative reward, the player needs to know the game by exploration, while guaranteeing current reward by exploitation.

How to incentivize exploration in RL has been a main focus in RL. Since RL is built on MAB, it is natural to extend MAB techniques to RL and UCB is such a success. UCB (Auer et al. (2002a)) motivates count-based exploration (Strehl and Littman, 2008) in RL and the subsequent Pseudo-Count exploration (Bellemare et al., 2016). New deep RL exploration algorithms have been recently proposed. Using deep neural networks to keep track of the $Q$-values by means of $Q$-networks in RL is called DQN (Mnih et al. (2013)). This combination of deep learning and RL has shown great success. $\epsilon$-greedy in Mnih et al. (2015) is a simple exploration technique using DQN. Besides $\epsilon$-greedy, intrinsic model exploration computes intrinsic rewards by focusing on experiences. Intrinsic rewards directly measure and incentivize exploration if added to extrinsic (actual) rewards of RL, e.g. DORA (Fox et al., 2018) and (Stadie et al., 2015). Random Network Distillation (RND) (Burda et al., 2018) is a more recent suggestion relying on a fixed target network. A drawback of RND is its local focus without global exploration.

In order to address weak points of these various exploration algorithms in the RL context, the notion of experts is natural and thus EXP-type MAB algorithms are appropriate. The allowance of arbitrary experts provides exploration for harder contextual bandits and hence providing exploration possibilities for RL. We develop an EXP4 exploration algorithm for RL that relies on several general experts. This is the first RL algorithm using several exploration experts enabling global exploration. Focusing on DQN, in the computational study we focus on two agents consisting of RND and $\epsilon$-greedy DQN.

We implement the RL EXP4 algorithm on the hard-to-explore RL game Montezuma's Revenge and compare it with the benchmark algorithm RND (Burda et al. (2018)). The numerical results show that the algorithm gains more exploration than RND and it gains the ability of global exploration by not getting stuck in local maximums of RND. Its total reward also increases with training. Overall, our algorithm improves exploration and exploitation on the benchmark game and demonstrates a learning process in RL.

Reward in RL in many cases is unbounded which relates to unbounded MAB rewards. There are three major versions of MAB: Adversarial, Stochastic, and herein introduced Gaussian. For adversarial MAB, rewards of the $K$ arms $r^t$ can be chosen arbitrarily by adversaries at step $t$. For stochastic MAB, the rewards at different steps are assumed to be i.i.d. and the rewards across arms are independent. It is assumed that $0 \leq r_i^t \leq 1$ for any arm $i$ and step $t$. For Gaussian MAB, rewards $r^t$ follow multi-variate normal $\mathcal{N}(\mu, \Sigma)$ with $\mu$ being the mean vector and $\Sigma$ the covariance matrix of the $K$ arms. Here the rewards are neither bounded, nor independent among the arms. For this reason the introduced Gaussian MAB reflects the RL setting and is the subject of our MAB analyses of EXP3.P.

EXP-type algorithms (Auer et al. (2002b)) are optimal in the two classical MABs. Auer et al. (2002b) show lower and upper bounds on regret of the order $O(\sqrt{T})$ for adversarial MAB and of the order $O(\log(T))$ for stochastic MAB. All of the proofs of these regret bounds by EXP-type algorithms are based on the bounded reward assumption, which does not hold for Gaussian MAB. Therefore, the regret bounds for Gaussian MAB with unbounded rewards studied herein are significantly different from prior works.

We show both lower and upper bounds on regret of Gaussian MAB under certain assumptions. Some analyses even hold for more generally distributed MAB. Upper bounds borrow some ideas from the analysis of the EXP3.P algorithm in Auer et al. (2002b) for bounded MAB to our unbounded MAB, while lower bounds are by our brand new construction of instances. Precisely, we derive lower bounds of order $\Omega(T)$ for certain fixed $T$ and upper bounds of order $O^*(\sqrt{T})$ for $T$ being large enough. The question of bounds for any value of $T$ remains open.

The main contributions of this work are as follows. On the analytical side we introduce Gaussian MAB with the unique aspect and challenge of unbounded rewards. We provide the very first regret lower bound in such a case by constructing a novel family of Gaussian bandits and we are able to analyze the EXP3.P algorithm for Gaussian MAB. Unbounded reward poses a non-trivial challenge in the analyses. We also provide the very first extension of EXP4 to RL exploration. We show its superior performance on two hard-to-explore RL games.

A literature review is provided in Section 2. Then in Section 3 we exhibit upper bounds for unbounded MAB of the EXP3.P algorithm and lower bounds, respectively. Section 4 discusses the EXP4 algorithm for RL exploration. Finally, in Section 5, we present numerical results related to the proposed algorithm.

## 2 LITERATURE REVIEW

The importance of exploration in RL is well understood. Count-based exploration in RL relies on UCB. Strehl and Littman (2008) develop Bellman value iteration $V(s) = \max_a \hat{R}(s, a) + \gamma E[V(s')] + \beta N(s, a)^{-\frac{1}{2}}$, where $N(s, a)$ is the number of visits to $(s, a)$ for state $s$ and action $a$. Value $N(s, a)^{-\frac{1}{2}}$ is positively correlated with curiosity of $(s, a)$ and encourages exploration. This method is limited to tableau model-based MDP for small state spaces, while Bellemare et al. (2016) introduce Pseudo-Count exploration for non-tableau MDP with density models.

In conjunction with DQN, $\epsilon$-greedy in Mnih et al. (2015) is a simple exploration technique using DQN. Besides $\epsilon$-greedy, intrinsic model exploration computes intrinsic rewards by the accuracy of a model trained on experiences. Intrinsic rewards directly measure and incentivize exploration if

added to extrinsic (actual) rewards of RL, e.g. DORA in Fox et al. (2018) and Stadie et al. (2015). Intrinsic rewards in Stadie et al. (2015) are defined as $e(s, a) = ||\sigma(s') - M_\phi(\sigma(s), a)||_2^2$ where $M_\phi$ is a parametric model, $s'$ is the next state and $\sigma$ is input extraction. Intrinsic reward $e(s, a)$ relies on stochastic transition from $s$ to $s'$ and brings noise to exploration. Random Network Distillation(RND) in Burda et al. (2018) addresses this by defining $e(s, a) = ||\hat{f}(s') - f(s')||_2^2$ where $\hat{f}$ is a parametric model and $f$ is a randomly initialized but fixed model. Here $e(s, a)$, independent of the transition, only depends on state $s'$ and drives RND to outperform other algorithms on Montezuma's Revenge. None of these algorithms use several experts which is a significant departure from our work.

In terms of MAB regret analyses focusing on EXP-type algorithms, Auer et al. (2002b) first introduce EXP3.P for bounded adversarial MAB and EXP4 for contextual bandits. Under the EXP3.P algorithm, an upper bound on regret of the order $O(\sqrt{T})$ is achieved, which has no gap with the lower bound and hence it establishes that EXP3.P is optimal. However these regret bounds are not applicable to Gaussian MAB since rewards can be infinite. Meanwhile for unbounded MAB, Srinivas et al. (2010) demonstrate a regret bound of order $O(\sqrt{T \cdot \gamma_T})$ for noisy Gaussian process bandits where a reward observation contains noise. The information gain $\gamma_T$ is not well-defined in a noiseless Gaussian setting. For noiseless Gaussian bandits, Grünewälder et al. (2010) show both the optimal lower and upper bounds on regret, but the regret definition is not consistent with the one used in Auer et al. (2002b). We establish a lower bound of the order $\Omega(T)$ for certain $T$ and an upper bound of the order $O^*(\sqrt{T})$ asymptotically on regret of unbounded noiseless Gaussian MAB following standard definitions of regret.

## 3 REGRET BOUNDS FOR GAUSSIAN MAB

For Gaussian MAB with time horizon $T$, at step $0 < t \leq T$ rewards $r^t$ follow multi-variate normal $\mathcal{N}(\mu, \Sigma)$ where $\mu = (\mu_1, \mu_2, \ldots, \mu_K)$ is the mean vector and $\Sigma = (a_{ij})_{i,j \in \{1, \ldots, K\}}$ is the covariance matrix of the $K$ arms. The player receives reward $y_t = r_{a_t}^t$ by pulling arm $a_t$. We use $R'_T = T \cdot \max_k \mu_k - \sum_t E[y_t]$ to denote pseudo regret called simply regret. (Note that the alternative definition of regret $R_T = \max_i \sum_{t=1}^T r_i^t - \sum_{t=1}^T y_t$ depends on realizations of rewards.)

### 3.1 LOWER BOUNDS ON REGRET

In this section we derive a lower bound for Gaussian and general MAB under an assumption. General MAB replaces Gaussian with a general distribution. The main technique is to construct instances or sub-classes that have certain regret, no matter what strategies are deployed. We need the following assumption or setting.

**Assumption 1** There are two types of arms with general $K$ with one type being superior ($S$ is the set of superior arms) and the other being inferior ($I$ is the set of inferior arms). Let $1 - q, q$ be the proportions of the superior and inferior arms, respectively which is known to the adversary and clearly $0 \leq q \leq 1$. The arms in $S$ are indistinguishable and so are those in $I$. The first pull of the player has two steps. In the first step the player selects an inferior or superior set of arms based on $P(S) = 1 - q$ and $P(I) = q$ and once a set is selected, the corresponding reward of an arm from the selected set is received.

An interesting special case of Assumption 1 is the case of two arms and $q = 1/2$. In this case, the player has no prior knowledge and in the first pull chooses an arm uniformly at random.

The lower bound is defined as $R_L(T) = \inf \sup R'_T$, where, first, $\inf$ is taken among all the strategies and then $\sup$ is among all Gaussian MAB. All proofs are in the Appendix.

The following is the main result with respect to lower bounds and it is based on inferior arms being distributed as $\mathcal{N}(0, 1)$ and superior as $\mathcal{N}(\mu, 1)$ with $\mu > 0$.

**Theorem 1.** *In Gaussian MAB under Assumption 1, for any $q \geq 1/3$ we have $R_L(T) \geq (q - \epsilon) \cdot \mu \cdot T$ where $\mu$ has to satisfy $G(q, \mu) < q$ with $\epsilon$ and $T$ determined by*

$$G(q, \mu) < \epsilon < q, \qquad T \leq \frac{\epsilon - G(q, \mu)}{(1 - q) \cdot \int \left| e^{-\frac{x^2}{2}} - e^{-\frac{(x-\mu)^2}{2}} \right|} + 2$$

*and $G(q, \mu) = \max \left\{ \int \left| q e^{-\frac{x^2}{2}} - (1-q) e^{-\frac{(x-\mu)^2}{2}} \right| dx, \int \left| (1-q) e^{-\frac{x^2}{2}} - q e^{-\frac{(x-\mu)^2}{2}} \right| dx \right\}.$*

To prove Theorem 1, we construct a special subset of Gaussian MAB with equal variances and zero covariances. On these instances we find a unique way to explicitly represent any policy. This builds a connection between abstract policies and this concrete mathematical representation. Then we show that pseudo regret $R'_T$ must be greater than certain values no matter what policies are deployed, which indicates a regret lower bound on these subset of instances.

The feasibility of the aforementioned conditions is established in the following theorem.

**Theorem 2.** *In Gaussian MAB under Assumption 1, for any $q \geq 1/3$, there exist $\mu$ and $\epsilon, \epsilon < \mu$ such that $R_L(T) \geq (q - \epsilon) \cdot \mu \cdot T$.*

The following result with two arms and equal probability in the first pull deals /with general probabilities. Even in the case of Gaussian MAB it is not a special case of Theorem 2 since it is stronger.

**Theorem 3.** *For general MAB under Assumption 1 with $K = 2, q = 1/2$, we have that $R_L(T) \geq \frac{T \cdot \mu}{4}$ holds for any distributions $f_0$ for the arms in $I$ and $f_1$ for the arms in $S$ with $\int |f_1 - f_0| > 0$ (possibly with unbounded support), for any $\mu > 0$ and $T$ satisfying $T \leq \frac{1}{2 \cdot \int |f_0 - f_1|} + 1$.*

The theorem establishes that for any fixed $\mu > 0$ there is a finite set of horizons $T$ and instances of Gaussian MAB so that no algorithm can achieve regret smaller than linear in $T$. Table 1 provides the values of the relationship between $\mu$ and largest $T$ in the Gaussian case where the inferior arms are distributed based on the standard normal and the superior arms have mean $\mu > 0$ and variance 1. For example, there is no way to attain regret lower than $T \cdot 10^{-4}/4$ for any $1 \leq T \leq 2501$. The function decreases very quickly.

Table 1: Upper bounds for $T$ as a function of $\mu$

| $\mu$ | $10^{-5}$ | $10^{-4}$ | $10^{-3}$ | $10^{-2}$ | $10^{-1}$ |
|---|---|---|---|---|---|
| Upper bound for $T$ | 25001 | 2501 | 251 | 26 | 3.5 |

The established lower bound result $R_L(T) \geq \Omega(T)$ is larger than known results of classical MAB. This is not surprising since the rewards in classical MAB are assumed to be bounded, while rewards in our setting follow an unbounded Gaussian distribution, which apparently increases regret.

Besides the known result $\Omega(\sqrt{T})$ of adversarial MAB and $\Omega(\log T)$ of stochastic MAB, for noisy Gaussian Process bandits, Srinivas et al. (2010) show $R_L(T) \leq \Omega(\sqrt{T \cdot \gamma_T})$. Our lower bound for Gaussian MAB is different from this lower bound. The information gain term $\gamma_T$ in noisy Gaussian bandits is not well-defined in Gaussian MAB and thus the two bounds are not comparable.

### 3.2 UPPER BOUNDS ON REGRET

In this section, we establish upper bounds for regret of Gaussian MAB by means of the EXP3.P algorithm (see Algorithm 1) from Auer et al. (2002b). We stress that rewards can be infinite, without the bounded assumption present in stochastic and adversarial MAB. We only consider non-degenerate Gaussian MAB where variance of each arm is strictly positive, i.e. $\min_i a_{ii} > 0$.

---

**Algorithm 1:** EXP3.P

---

Initialization: Weights $w_i(1) = \exp\left(\frac{\alpha \delta}{3}\sqrt{\frac{T}{K}}\right), i \in \{1, 2, \ldots, K\}$ for $\alpha > 0$ and $\delta \in (0, 1)$;

**for** $t = 1, 2, \ldots, T$ **do**

    **for** $i = 1, 2, \ldots, K$ **do**

        $p_i(t) = (1 - \delta)\frac{w_i(t)}{\sum_{j=1}^{K} w_j(t)} + \frac{\delta}{K}$

    **end**

    Choose $i_t$ randomly according to the distribution $p_1(t), \ldots, p_K(t)$;

    Receive reward $r_{i_t}(t)$;

    **for** $j = 1, \ldots, K$ **do**

        $\hat{x}_j(t) = \frac{r_j(t)}{p_j(t)} \cdot \mathbb{1}_{j=i_t}, \quad w_j(t+1) = w_j(t) \exp\frac{\delta}{3K}\left(\hat{x}_j(t) + \frac{\alpha}{p_j(t)\sqrt{KT}}\right)$

    **end**

**end**

---

Formally, we provide analyses for upper bounds on $R_T$ with high probability, on $E[R_T]$ and on $R'_T$. In Auer et al. (2002b) EXP3.P is studied to yield a bound on regret $R_T$ with high probability in the bounded MAB setting. As part of our contributions, we show that EXP3.P regret is of the order $O^*(\sqrt{T})$ in the unbounded Gaussian MAB in the case of $R_T$ with high probability, $E[R_T]$ and $R'_T$. The results are summarized as follows. The density of $\mathcal{N}(\mu, \Sigma)$ is denoted by $f$.

**Theorem 4.** *For Gaussian MAB, any time horizon $T$, for any $0 < \eta < 1$, EXP3.P has regret*

$$R_T \leq 4\Delta(\eta) \cdot (\sqrt{KT \log(\frac{KT}{\delta})} + 4\sqrt{\frac{5}{3} KT \log K} + 8 \log(\frac{KT}{\delta})) \text{ with probability } (1-\delta) \cdot (1-\eta)^T$$

*where $\Delta(\eta)$ is determined by $\int_{-\Delta}^{\Delta} \ldots \int_{-\Delta}^{\Delta} f(x_1, \ldots, x_K) \, dx_1 \ldots dx_K = 1 - \eta$.*

In the proof of Theorem 4, we first perform truncation of the rewards of Gaussian MAB by dividing the rewards to a bounded part and unbounded tail throughout the game. For the bounded part, we directly borrow the regret upper bound of EXP3.P in Auer et al. (2002b) and conclude with the regret upper bound of order $O(\Delta(\eta)\sqrt{T})$. Since a Gaussian distribution is a light-tailed distribution we can control the probability of tail shrinking which leads to the overall result.

The dependence of the bound on $\Delta$ can be removed by considering large enough $T$ as stated next.

**Theorem 5.** *For Gaussian MAB, and any $a > 2$, $0 < \delta < 1$, EXP3.P has regret*

$$R_T \leq \log(1/\delta)O^*(\sqrt{T}) \text{ with probability } (1-\delta) \cdot (1 - \frac{1}{T^a})^T.$$

The constant behind $O^*$ depends on $K, a, \mu$ and $\Sigma$.

The above theorems deal with $R_T$ but the aforementioned lower bounds are with respect to pseudo regret. To complete the analysis of Gaussian MAB, it is desirable to have an upper bound on pseudo regret which is established next. It is easy to verify by the Jensen's inequality that $R'_T \leq E[R_T]$ and thus it suffices to obtain an upper bound on $E[R_T]$.

For adversarial and stochastic MAB, the upper bound for $E[R_T]$ is of the same order as $R_T$ which follows by a simple argument. For Gaussian MAB, establishing an upper bound on $E[R_T]$ or $R'_T$ based on $R_T$ requires more work. We show an upper bound on $E[R_T]$ by using select inequalities, limit theories, and Randemacher complexity. To this end, the main result reads as follows.

**Theorem 6.** *The regret of EXP3.P in Gaussian MAB satisfies*

$$R'_T \leq E[R_T] \leq O^*(\sqrt{T}).$$

All these three theorems also hold for sub-Gaussian MAB, which is defined by replacing Gaussian with sub-Gaussian. This generalization is straightforward and it is directly shown in the proof of Gaussian MAB in Appendix. Optimal upper bounds for adversarial MAB and noisy Gaussian Process bandits are of the same order as our upper bound. Auer et al. (2002b) derive an upper bound of the same order $O(\sqrt{T})$ as the lower bound for adversarial MAB. For noisy Gaussian Process bandits, there is also no gap between its upper and lower bounds.

Our upper bound of the order $O^*(\sqrt{T})$ is of the same order as the one for bounded MAB. In our case the upper bound result $O^*(\sqrt{T})$ holds for large enough $T$ which is hidden behind $O^*$ while the linear lower bounds is valid only for small values of $T$. This illustrates the rationality of the lower bound of $O(T)$ and the upper bound of order $O^*(\sqrt{T})$.

## 4 EXP4 ALGORITHM FOR RL

EXP4 has shown great success in contextual bandits. Therefore, in this section, we extend EXP4 to RL and develop EXP4-RL illustrated in Algorithm 2.

The player has experts that are represented by deep $Q$-networks trained by RL algorithms (there is a one to one correspondence between the experts and $Q$-networks). Each expert also has a trust coefficient. Trust coefficients are also updated exponentially based on the reward estimates as in EXP4. At each step of one episode, the player samples an expert ($Q$-network) with probability that is proportional to the weighted average of expert's trust coefficients. Then $\epsilon$-greedy DQN is applied on the chosen $Q$-network. Here different from EXP4, the player needs to store all the interaction tuples

in experience buffer since RL is a MDP. After one episode, the player trains all $Q$-networks with the experience buffer and uses the trained networks as experts for the next episode.

---

**Algorithm 2:** EXP4-RL

---

Initialization: Trust coefficients $w_k = 1$ for any $k \in \{1, \dots, E\}$, $E =$ number of experts ($Q$-networks), $K =$ number of actions, $\Delta, \epsilon, \eta > 0$ and temperature $z, \tau > 0$, $n_r = -\infty$ (an upper bound on reward);

**while** *True* **do**

    Initialize episode by setting $s_0$;

    **for** $i = 1, 2, \dots, T$*(length of episode)* **do**

        Observe state $s_i$;

        Let probability of $Q_k$-network be $\rho_k = (1 - \eta)\frac{w_k}{\sum_{k=1}^{E} w_k} + \frac{\eta}{E}$;

        Sample network $\bar{k}$ according to $\{\rho_k\}_k$;

        For $Q_{\bar{k}}$-network, use $\epsilon$-greedy to sample an action

$$a^* = argmax_a Q_{\bar{k}}(s_i, a), \qquad \pi_j = (1-\epsilon)\cdot\mathbb{1}_{j=a^*} + \frac{\epsilon}{K-1}\cdot\mathbb{1}_{j\neq a^*} \qquad j \in \{1, 2, \dots, K\}$$

        Sample action $a_i$ based on $\pi$;

        Interact with the environment to receive reward $r_i$ and next state $s_{i+1}$;

        $n_r = \max\{r_i, n_r\}$;

        Update the trust coefficient $w_k$ of each $Q_k$-network as follows:

$$P_k = \epsilon\text{-greedy}(Q_k), \hat{x}_{kj} = 1 - \frac{\mathbb{1}_{j=a}}{P_{kj} + \Delta}(n_r - r_i), j \in 1, 2, \dots, K, y_k = E[\hat{x}_{kj}], w_k = w_k\cdot e^{\frac{y_k}{z}}$$

        Store $(s_i, a_i, r_i, s_{i+1})$ in experience replay buffer $B$;

    **end**

    Update each expert's $Q_k$-network from buffer $B$;

**end**

---

The basic idea is the same as EXP4 by using the experts that give advice vectors with deep $Q$-networks. It is a combination of deep neural networks with EXP4 updates. From a different perspective, we can also view it as an ensemble in classification (Xia et al. (2011)), by treating $Q$-networks as ensembles in RL, instead of classification algorithms. While $Q$-networks do not necessarily have to be experts, i.e., other experts can be used, these are natural in a DQN framework.

In our implementation and experiments we use two experts, thus $E = 2$ with two $Q$-networks. The first one is based on RND (Burda et al. (2018)) while the second one is a simple DQN. To this end, in the algorithm before storing to the buffer, we also record $c_r^i = ||\hat{f}(s_i) - f(s_i)||^2$, the RND intrinsic reward as in Burda et al. (2018). This value is then added to the 4-tuple pushed to $B$. When updating $Q_1$ corresponding to RND at the end of an iteration in the algorithm, by using $r_j + c_r^j$ we modify the $Q_1$-network and by using $c_r^j$ an update to $\hat{f}$ is executed. Network $Q_2$ pertaining to $\epsilon$-greedy is updated directly by using $r_j$.

Intuitively, Algorithm 2 circumvents this drawback with the total exploration guided by two experts with EXP4 updated trust coefficients. When the RND expert drives high exploration, its trust coefficient leads to a high total exploration. When it has low exploration, the second expert DQN should have a high one and it incentivizes the total exploration accordingly. Trust coefficients are updated by reward estimates iteratively as in EXP4, so they keep track of the long-term performance of experts and then guide the total exploration globally. These dynamics of EXP4 combined with intrinsic rewards guarantees global exploration. The experimental results exhibited in the next section verify this intuition regarding exploration behind Algorithm 2.

We point out that potentially more general RL algorithms based on $Q$-factors can be used, e.g., boostrapped DQN (Osband et al. (2016)), random prioritized DQN (Osband et al. (2018)) or adaptive $\epsilon$-greedy VDBE (Tokic (2010)) are a possibility. Furthermore, experts in EXP4 can even be policy networks trained by PPO (Schulman et al. (2017)) instead of DQN for exploration. These possibilities demonstrate the flexibility of the EXP4-RL algorithm.

## 5 COMPUTATIONAL STUDY

As a numerical demonstration of the superior performance and exploration incentive of Algorithm 2, we show the improvements on baselines on two hard-to-explore RL games, Mountain Car and Montezuma's Revenge. More precisely, we present that the real reward on Mountain Car improves significantly by Algorithm 2 in Section 5.1. Then we implement Algorithm 2 on Montezuma's Revenge and show the growing and remarkable improvement of exploration in Section 5.2.

Intrinsic reward $c_r^i = ||\hat{f}(s_i) - f(s_i)||^2$ given by intrinsic model $\hat{f}$ represents the exploration of RND in Burda et al. (2018) as introduced in Sections 2 and 4. We use the same criterion for evaluating exploration performance of our algorithm and RND herein. RND incentivizes local exploration with the single step intrinsic reward but with the absence of global exploration.

### 5.1 MOUNTAIN CAR

In this part, we summarize the experimental results of Algorithm 2 on Mountain Car, a classical control RL game. This game has very sparse positive rewards, which brings the necessity and hardness of exploration. Blog post (Rivlin (2019)) shows that RND based on DQN improves the performance of traditional DQN, since RND has intrinsic reward to incentivize exploration. We use RND on DQN from Rivlin (2019) as the baseline and show the real reward improvement of Algorithm 2, which supports the intuition and superiority of the algorithm.

The comparison between Algorithm 2 and RND is presented in Figure 1. Here the x-axis is the epoch number and the y-axis is the cumulative reward of that epoch. Figure 1a shows the raw data comparison between EXP4-RL and RND. We observe that though at first RND has several spikes exceeding those of EXP4-RL, EXP4-RL has much higher rewards than RND after 300 epochs. Overall, the relative difference of areas under the curve (AUC) is 4.9% for EXP4-RL over RND, which indicates the significant improvement of our algorithm. This improvement is better illustrated in Figure 1b with the smoothed reward values. Here there is a notable difference between EXP4-RL and RND. Note that the maximum reward hit by EXP4-RL is $-86$ and the one by RND is $-118$, which additionally demonstrates our improvement on RND.

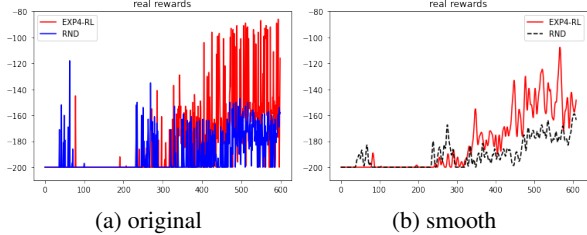

(a) original      (b) smooth

Figure 1: The performance of Algorithm 2 and RND measured by the epoch-wise reward on Mountain Car, with the left one being the original data and the right being the smoothed reward values.

We conclude that Algorithm 2 performs better than the RND baseline and that the improvement increases at the later training stage. Exploration brought by Algorithm 2 gains real reward on this hard-to-explore Mountain Car, compared to the RND counterpart (without the DQN expert). The power of our algorithm can be enhanced by adopting more complex experts, not limited to only DQN.

### 5.2 MONTEZUMA'S REVENGE AND PURE EXPLORATION SETTING

In this section, we show the experimental details of Algorithm 2 on Montezuma's Revenge, another notoriously hard-to-explore RL game. The benchmark on Montezuma's Revenge is RND based on DQN which achieves a reward of zero in our environment (the PPO algorithm reported in Burda et al. (2018) has reward 8,000 with many more computing resources; we ran the PPO-based RND with 10 parallel environments and 800 epochs to observe that the reward is also 0), which indicates that DQN has room for improvement regarding exploration.

To this end, we first implement the DQN-version RND (called simply RND hereafter) on Montezuma's Revenge as our benchmark by replacing the PPO with DQN. Then we implement Algorithm 2 with two experts as aforementioned. Our computing environment allows at most 10 parallel environments. In subsequent figures the x-axis always corresponds to the number of epochs. RND update probability is the proportion of experience that are used for training the intrinsic model $\hat{f}$ (Burda et al., 2018).

A comparison between Algorithm 2 (EXP4-RL) and RND without parallel environments (the update probability is 100% since it is a single environment) is shown in Figure 2 with the emphasis on exploration by means of the intrinsic reward. We use 3 different numbers of burn-in periods (58, 68, 167 burn-in epochs) to remove the initial training steps, which is common in Gibbs sampling. Overall EXP4-RL outperforms RND with many significant spikes in the intrinsic rewards. The larger the number of burn-in periods is, the more significant is the dominance of EXP4-RL over RND. EXP4-RL has much higher exploration than RND at some epochs and stays close to RND at other epochs. At some epochs, EXP4-RL even has 6 times higher exploration. The relative difference in the areas under the curves are 6.9%, 17.0%, 146.0%, respectively, which quantifies the much better performance of EXP4-RL.

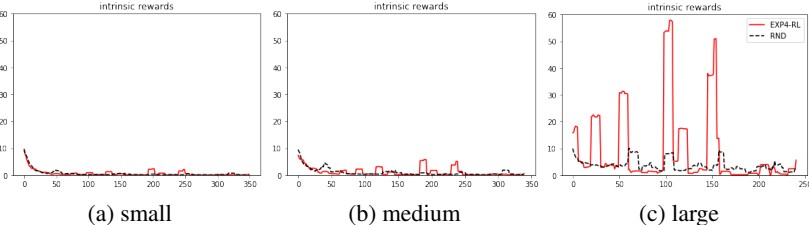

(a) small            (b) medium            (c) large

Figure 2: The performance of Algorithm 2 and RND measured by intrinsic reward without parallel environments with three different burn-in periods

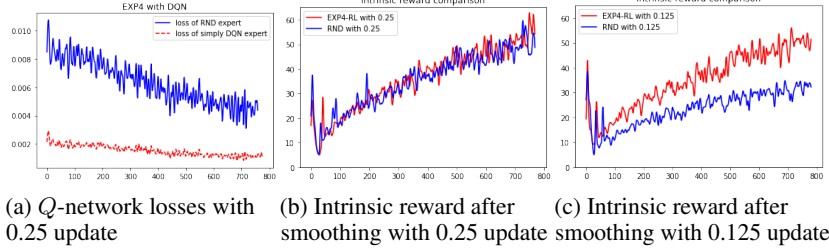

(a) $Q$-network losses with 0.25 update     (b) Intrinsic reward after smoothing with 0.25 update     (c) Intrinsic reward after smoothing with 0.125 update

Figure 3: The performance of Algorithm 2 and RND with 10 parallel environments and with RND update probability 0.25 and 0.125, measured by loss and intrinsic reward.

We next compare EXP4-RL and RND with 10 parallel environments and different RND update probabilities in Figure 3. The experiences are generated by the 10 parallel environments.

Figure 3a shows that both experts in EXP4-RL are learning with decreasing losses of their $Q$-networks. The drop is steeper for the RND expert but it starts with a higher loss. With RND update probability 0.25 in Figure 3b we observe that EXP4-RL and RND are very close when RND exhibits high exploration. When RND is at its local minima, EXP4-RL outperforms it. Usually these local minima are driven by sticking to local maxima and then training the model intensively at local maxima, typical of the RND local exploration behavior. EXP4-RL improves on RND as training progresses, e.g. the improvement after 550 epochs is higher than the one between epochs 250 and 550. In terms for AUC, this is expressed by 1.6% and 3.5%, respectively. Overall, EXP4-RL improves RND local minima of exploration, keeps high exploration of RND and induces a smoother global exploration.

With the update probability of 0.125 in Figure 3c, EXP4-RL almost always outperforms RND with a notable difference. The improvement also increases with epochs and is dramatically larger at RND's local minima. These local minima appear more frequently in training of RND, so our improvement is more significant as well as crucial. The relative AUC improvement is 49.4%. The excellent performance in Figure 3c additionally shows that EXP4-RL improves RND with global exploration by improving local minima of RND or not staying at local maxima.

Overall, with either 0.25 or 0.125, EXP4-RL incentivizes global exploration on RND by not getting stuck in local exploration maxima and outperforms RND exploration aggressively. With 0.125 the improvement with respect to RND is more significant and steady. These experimental evidence verifies our intuition behind EXP4-RL and provides excellent support for it. With experts being more advanced RL exploration algorithms, e.g. DORA, EXP4-RL can bring additional possibilities.

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

## A    DETAILS ABOUT NUMERICAL EXPERIMENTS

### A.1    MOUNTAIN CAR

For the Mountain Car experiment, we use the Adam optimizer with the $2 \cdot 10^{-4}$ learning rate. The batch size for updating models is 64 with the replay buffer size of 10,000. The remaining parameters are as follows: the discount factor for the $Q$-networks is 0.95, the temperature parameter $\tau$ is 0.1, $\eta$ is 0.05, and $\epsilon$ is decaying exponentially with respect to the number of steps with maximum 0.9 and minimum 0.05. The length of one epoch is 200 steps. The target networks load the weights and biases of the trained networks every 400 steps. Since a reward upper bound is known in advance, we use $n_r = 1$.

We next introduce the structure of neural networks that are used in the experiment. The neural networks of both experts are linear. For the RND expert, it has the input layer with 2 input neurons, followed by a hidden layer with 64 neurons, and then a two-headed output layer. The first output layer represents the $Q$ values with 64 hidden neurons as input and the number of actions output neurons, while the second output layer corresponds to the intrinsic values, with 1 output neuron. For the DQN expert, the only difference lies in the absence of the second output layer.

### A.2    MONTEZUMA'S REVENGE

For the Montezuma's Revenge experiment, we use the Adam optimizer with the $10^{-5}$ learning rate. The other parameters read: the mini batch size is 4, replay buffer size is 1,000, the discount factor for the $Q$-networks is 0.999 and the same valus is used for the intrinsic value head, the temperature parameter $\tau$ is 0.1, $\eta$ is 0.05, and $\epsilon$ is increasing exponentially with minimum 0.05 and maximum 0.9. The length of one epoch is 100 steps. Target networks are updated every 300 steps. Pre-normalization is 50 epochs and the weights for intrinsic and extrinsic values in the first network are 1 and 2, respectively. The upper bound on reward is set to be constant $n_r = 1$.

For the structure of nerual networks, we use CNN architectures since we are dealing with videos. More precisely, for the $Q$-network of the DQN expert in EXP4-RL and the predictor network $\hat{f}$ for computing the intrinsic rewards, we use Alexnet (Krizhevsky et al. (2012)) pretrained on ImageNet (Deng et al. (2009)). The number of output neurons of the final layer is 18, the number of actions in Montezuma. For the RND baseline and RND expert in EXP4-RL, we customize the $Q$-network with different linear layers while keeping all the layers except the final layer of pretrained Alexnet. Here we have two final linear layers representing two value heads, the extrinsic value head and the intrinsic value head. The number of output neurons in the first value head is again 18, while the second value head is with 1 output neuron.

More details about the setup of the experiment on Montezuma's Revenge are elaborated as follows. The experiment of RND with PPO in Burda et al. (2018) uses many more resources, such as 1024 parallel environments and runs 30,000 epochs for each environment. Parallel environments generate experiences simultaneously and store them in the replay buffer. Our computing environment allows at most 10 parallel environments. For the DQN-version of RND (called simply RND hereafter), we use the same settings as Burda et al. (2018), such as observation normalization, intrinsic reward normalization and random initialization. RND update probability is the proportion of experience in the replay buffer that are used for training the intrinsic model $\hat{f}$ in RND (Burda et al., 2018). Here in our experiment, we compare the performance under 0.125 and 0.25 RND update probability.

## B    PROOF OF RESULTS IN SECTION 3.1

For brevity, we define $n = T - 1$.

We start by showing the following proposition that is used in the proofs.

**Proposition 1.** *Let $G(q, \mu), q,$ and $\mu$ be defined as in Theorem 1. Then for any $q \geq 1/3$, there exists a $\mu$ that satisfies the constraint $G(q, \mu) < q$.*

*Proof.* Let us denote $G_1 = \int |qf_0(x) - (1-q)f_1(x)| \, dx, G_2 = \int |(1-q)f_0(x) - qf_1(x)| \, dx$. Then we have

$$G_1(q,\mu) = \int |qf_0(x) - (1-q)f_1(x)| \, dx$$

$$= \int (qf_0(x) - (1-q)f_1(x)) \, \mathbb{1}_{qf_0(x)>(1-q)f_1(x)} dx$$

$$+ \int (-qf_0(x) + (1-q)f_1(x)) \, \mathbb{1}_{qf_0(x)<(1-q)f_1(x)} dx$$

$$= \int (qf_0(x) - (1-q)f_1(x)) \, \mathbb{1}_{x<g(\mu)} dx + \int (-qf_0(x) + (1-q)f_1(x)) \, \mathbb{1}_{x>g(\mu)} dx$$

$$= \frac{1}{\sqrt{2\pi}} \left[ \int_{-\infty}^{g(\mu)} \left( qe^{-\frac{x^2}{2}} - (1-q)e^{-\frac{(x-\mu)^2}{2}} \right) dx + \int_{g(\mu)}^{\infty} \left( -qe^{-\frac{x^2}{2}} + (1-q)e^{-\frac{(x-\mu)^2}{2}} \right) dx \right]$$

$$= \frac{1}{\sqrt{2\pi}} \left[ q \int_{-g(\mu)}^{g(\mu)} e^{-\frac{x^2}{2}} - (1-q) \int_{-g(\mu)+\mu}^{g(\mu)-\mu} e^{-\frac{x^2}{2}} \right]$$

where $g(\mu) = \frac{1}{2} \cdot \mu - \frac{\log(\frac{1-q}{q})}{\mu}$. Similarly we get

$$G_2(q,\mu) = \frac{1}{\sqrt{2\pi}} \left[ (1-q) \int_{-g(\mu)}^{g(\mu)} e^{-\frac{x^2}{2}} - q \int_{-g(\mu)+\mu}^{g(\mu)-\mu} e^{-\frac{x^2}{2}} \right].$$

It is easy to establish continuity of $G_1(q,\mu)$ and $G_2(q,\mu)$ on $[0,\infty)$, as well as the continuity of $G(q,\mu)$. Indeed, we have

$$G(q,\mu) = \begin{cases} |1 - 2q| & \mu = 0 \\ \max(q, 1-q) & \mu \to \infty . \end{cases}$$

Since $q \geq \frac{1}{3}$, then $|1 - 2q| < q$. From continuity of $G(q,\mu)$, there exists $\mu_0 > 0$ such that $G(q,\mu) < q$ for any $\mu \leq \mu_0$. $\square$

*Proof of Theorem 1.* As in Assumption 1, let the inferior arm set be $I$ and the superior one be $S$, respectively, $P(I) = q$ and $P(S) = 1 - q$. Arms in $I$ follow $f_0(x) = \mathcal{N}(0,1)$ and arms in $S$ follow $f_1(x) = \mathcal{N}(\mu,1)$ where $\mu > 0$. According to Assumption 1, at the first step the player pulls an arm from either $I$ or $S$ and receives reward $y_1$. At time step $i > 1$, the reward is $y_i$ and let $b_i$ represent a policy of the player. We can always define $b_i$ as

$$b_i = \begin{cases} 1 & \text{if the chosen arm at step } i \text{ is not in the same arm set as the initial arm,} \\ 0 & \text{otherwise.} \end{cases}$$

Let $a_i \in \{0, 1\}$ be the actual arm played at step $i$. It suffices to only specify $a_i$ is in arm set $I$ ($a_i = 0$) or $S$ ($a_i = 1$) since the arms in $I$ and $S$ are identical. The connection between $a_i$ and $b_i$ is explicitly given by $b_i = |a_i - a_1|$. By Assumption 1, it is easy to argue that $b_i = S'_i(y_1, y_2, ..., y_{i-1})$ for a set of functions $S'_2, S'_3, \ldots, S'_n, S'_{n+1}$. We proceed with the following lemma.

**Lemma 1.** *Let the rewards of the arms in set $I$ follow any $L_1$ distribution $f_0(x)$ and in set $S$ follow any $L_1$ distribution $f_1(x)$ where the means satisfy $\mu(f_1) > \mu(f_0)$. Let $B$ be the number of arms played in the game in set $S$. Let us assume the player meets Assumption 1. Then no matter what strategy the player takes, we have*

$$\left| \frac{E[B] - (1-q) \cdot (n+1)}{n+1} \right| \leq \epsilon$$

*where $\epsilon, T, f_0, f_1$ satisfy*

$$G(q, f_0, f_1) + (1-q)(n-1) \int |f_0(x) - f_1(x)| \leq \epsilon,$$

$$G(q, f_0, f_1) = \max \left\{ \int |q f_0(x) - (1 - q) f_1(x)| \, dx, \int |(1 - q) f_0(x) - q f_1(x)| \, dx \right\}.$$

*Proof.* We have

$$E[B] = \int (a_1 + a_2 + \cdots + a_{n+1}) f_{a_1}(y_1) f_{a_2}(y_2) \ldots f_{a_n}(y_n) \, dy_1 dy_2 \ldots dy_n.$$

If $a_1 = 0$, then $a_i = b_i$ and

$$E[B|a_1 = 0] = \int (0 + b_2(y_{1:1}) + \ldots + b_{n+1}(y_{1:n})) f_0(y_1) f_{b_2}(y_2) \ldots f_{b_n}(y_n) \, dy_1 dy_2 \ldots dy_n.$$

If $a_1 = 1$, then $1 - a_i = b_i$ and

$$E[B|a_1 = 1] = \int (1 + 1 - b_2(y_{1:1}) + \cdots + 1 - b_{n+1}(y_{1:n})) f_1(y_1) \ldots f_{1-b_n}(y_n) \, dy_1 dy_2 \ldots dy_n.$$

This gives us

$$\begin{aligned}
E[B] &= q \cdot E[B|a_1 = 0] + (1 - q) \cdot E[B|a_1 = 1] \\
&= (1 - q)(n + 1) \\
&\quad + \int (b_2 + \cdots + b_{n+1}) \cdot (q \cdot f_0(y_1) \ldots f_{b_n}(y_n) - (1 - q) \cdot f_1(y_1) \ldots f_{1-b_n}(y_n)) \, dy_1 dy_2 \ldots dy_n.
\end{aligned}$$

By defining $b_1 = 0$, we have

$$\begin{aligned}
E[B] &= (1 - q) \cdot (n + 1) + \\
&\int (b_2 + \cdots + b_{n+1}) (q \cdot f_{b_1}(y_1) \ldots f_{b_n}(y_n) - (1 - q) \cdot f_{1-b_1}(y_1) \ldots f_{1-b_n}(y_n)) \, dy_1 dy_2 \ldots dy_n.
\end{aligned}$$

For any $1 \leq m \leq n$ we also derive

$$\begin{aligned}
&\int \left| \prod_{i=1}^{m} f_{b_i}(y_i) - \prod_{i=1}^{m} f_{1-b_i}(y_i) \right| dy_1 dy_2 \ldots dy_m \\
&\leq \int \prod_{i=1}^{m-1} f_{b_i}(y_i) |f_{b_m}(y_m) - f_{1-b_m}(y_n)| \, dy_1 dy_2 \ldots dy_m + \\
&\quad \int \left| \prod_{i=1}^{m-1} f_{b_i}(y_i) - \prod_{i=1}^{m-1} f_{1-b_i}(y_i) \right| f_{1-b_m}(y_m) \, dy_1 dy_2 \ldots dy_m \\
&\leq \int |f_0(x) - f_1(x)| \, dx + \int \left| \prod_{i=1}^{m-1} f_{b_i}(y_i) - \prod_{i=1}^{m-1} f_{1-b_i}(y_i) \right| f_{1-b_m}(y_m) \, dy_1 dy_2 \ldots dy_m \qquad (1) \\
&= \int |f_0(x) - f_1(x)| \, dx + \int \left| \prod_{i=1}^{m-1} f_{b_i}(y_i) - \prod_{i=1}^{m-1} f_{1-b_i}(y_i) \right| dy_1 dy_2 \ldots dy_{m-1} \\
&\leq 2 \cdot \int |f_0(x) - f_1(x)| \, dx + \int \left| \prod_{i=1}^{m-2} f_{b_i}(y_i) - \prod_{i=1}^{m-2} f_{1-b_i}(y_i) \right| dy_1 dy_2 \ldots dy_{m-2} \\
&\leq m \int |f_0(x) - f_1(x)|.
\end{aligned}$$

This provides

$$\left| \frac{E[B] - (1-q) \cdot (n+1)}{n+1} \right|$$

$$\leq \int \left| q \cdot \prod_{i=1}^{n} f_{b_i}(y_i) - (1-q) \cdot \prod_{i=1}^{n} f_{1-b_i}(y_i) \right| dy_1 dy_2 \ldots dy_n$$

$$\leq \int \prod_{i=1}^{n-1} f_{b_i}(y_i) \left| q \cdot f_{b_n}(y_n) - (1-q) \cdot f_{1-b_n}(y_n) \right| dy_1 dy_2 \ldots dy_n +$$

$$\int \left| (1-q) \cdot \prod_{i=1}^{n-1} f_{b_i}(y_i) - (1-q) \cdot \prod_{i=1}^{n-1} f_{1-b_i}(y_i) \right| f_{1-b_n}(y_n) \, dy_1 dy_2 \ldots dy_n$$

$$\leq \max \left\{ \int |q \cdot f_0(x) - (1-q) \cdot f_1(x)| \, dx, \int |(1-q) \cdot f_0(x) - q \cdot f_1(x)| \, dx \right\} +$$

$$(1-q) \cdot \int \left| \prod_{i=1}^{n-1} f_{b_i}(y_i) - \prod_{i=1}^{n-1} f_{1-b_i}(y_i) \right| dy_1 dy_2 \ldots dy_{n-1}$$

$$\leq \max \left\{ \int |q \cdot f_0(x) - (1-q) \cdot f_1(x)| \, dx, \int |(1-q) \cdot f_0(x) - q \cdot f_1(x)| \, dx \right\} +$$

$$(1-q) \cdot (n-1) \cdot \int |f_0(x) - f_1(x)|,$$

where the last inequality follows from (1).The statement of the lemma now follows. $\qquad\square$

According to Proposition 1, there is such $\mu$ satisfying the constraint $G(q, \mu) < q$. Note that $G(q, \mu) = G(q, f_0, f_1)$. Then we can choose $\epsilon$ to be any quantity such that $G(q, \mu) < \epsilon < q$. Finally, there is $T$ satisfying $T \leq \frac{\epsilon - G(q,\mu)}{(1-q) \cdot \int |f_0(x) - f_1(x)|} + 2$ that gives us

$$G(q, \mu) + (1-q)(T-2) \int |f_0(x) - f_1(x)| \leq \epsilon.$$

By choosing $\epsilon, T, \mu$ as above, by Lemma 1 we have

$$\left| \frac{E[B] - (1-q) \cdot T}{T} \right| < \epsilon,$$

which is equivalent to $E[B] < (1 - q + \epsilon) \cdot T$. Therefore, regret $R'_T$ satisfies, with $A$ being the number of arm pulls from $I$, inequality

$$R'_T = \sum_t \max_k(\mu_k) - \sum_t E[y_t] = T\mu - \sum_t E[y_t] = T\mu - (E[B] \cdot \mu + E[A] \cdot 0)$$
$$\geq T\mu - (1 - q + \epsilon)\mu T = (q - \epsilon)\mu T.$$

This yields $R_T^L = \inf \sup R'_T \geq (q - \epsilon) \cdot \mu T.$ $\qquad\square$

Theorem 2 follows from Theorem 1 and Proposition 1.

*Proof of Theorem 3.* The assumption here is the special case of Assumption 1 where there are two arms and $q = 1/2$. Set $I$ follows $f_0$ and $S$ follows $f_1$ where $\mu(f_0) < \mu(f_1)$.

In the same was as in the proof of Theorem 1 we obtain

$$R_L(T) \geq \left( \tfrac{1}{2} - \epsilon \right) \cdot T \cdot \mu$$

under the constraint that $n/2 \cdot \int |f_0 - f_1| = n/2 \cdot \mathrm{TV}(f_0, f_1) < \epsilon$ where TV stands for total variation. Here we use $G(1/2, \mu) = 1/2 \cdot \mathrm{TV}(f_0, f_1)$. Setting $\epsilon = 1/4$ yields the statement. $\qquad\square$

In the Gaussian case it turns out that $\epsilon = 1/4$ yields the highest bound. For total variation of Gaussian variables $N(\mu_1, \sigma_1^2)$ and $N(\mu_2, \sigma_2^2)$, Devroye et al. (2018) show that

$$\text{TV}\left(\mathcal{N}\left(\mu_1, \sigma_1^2\right), \mathcal{N}\left(\mu_2, \sigma_2^2\right)\right) \leq \frac{3|\sigma_1^2 - \sigma_2^2|}{2\sigma_1^2} + \frac{|\mu_1 - \mu_2|}{2\sigma_1},$$

which in our case yields $TV \leq \frac{\mu}{2}$. From this we obtain $\mu \cdot T \geq \epsilon$ and in turn $R_T^L \geq \epsilon \cdot (\frac{1}{2} - \epsilon)$. The maximum of the right-hand side is obtained at $\epsilon = \frac{1}{4}$. This justifies the choice of $\epsilon$ in the proof of 3.

## C   PROOF OF RESULTS IN SECTION 3.2

### C.1   PROOF FOR THEOREM 4

*Proof.* Since the rewards can be unbounded in our setting, we consider truncating the reward with any $\Delta > 0$ for any arm $i$ by $r_i^t = \bar{r}_i^t + \hat{r}_i^t$ where

$$\bar{r}_i^t = r_i^t \cdot \mathbb{1}_{(-\Delta \leq r_i^t \leq \Delta)}, \hat{r}_i^t = r_i^t \cdot \mathbb{1}_{(|r_i^t| > \Delta)}.$$

Then for any parameter $0 < \eta < 1$, we choose such $\Delta$ that satisfies

$$P(r_i^t = \bar{r}_i^t, i \leq K) = P(-\Delta \leq r_1^t \leq \Delta, \ldots, -\Delta \leq r_K^t \leq \Delta)$$
$$= \int_{-\Delta}^{\Delta} \int_{-\Delta}^{\Delta} \ldots \int_{-\Delta}^{\Delta} f(x_1, \ldots, x_K) dx_1 \ldots dx_K \geq 1 - \eta. \tag{2}$$

The existence of such $\Delta = \Delta(\eta)$ follows from elementary calculus.

Let $A = \{|r_i^t| \leq \Delta$ for every $i \leq K, t \leq T\}$. Then the probability of this event is

$$P(A) = P(r_i^t = \bar{r}_i^t, i \leq K, t \leq T) \geq (1 - \eta)^T.$$

With probability $(1 - \eta)^T$, the rewards of the player are bounded in $[-\Delta, \Delta]$ throughout the game. Then $R_T^B = \sum_{t=1}^{T}(\max_i \bar{r}_i^t - \bar{r}_i^i) \leq T \cdot \Delta - \sum_{t=1}^{T} r_t = R_T^B$ is the regret under event $A$, i.e. $R_T = R_T^B$ with probability $(1 - \eta)^T$. For the EXP3.P algorithm and $R_T^B$, for every $\delta > 0$, according to Auer et al. (2002b) we have

$$R_T^B \leq 4\Delta \left( \sqrt{KT \log(\frac{KT}{\delta})} + 4\sqrt{\frac{5}{3}KT \log K} + 8 \log(\frac{KT}{\delta}) \right) \text{ with probability } 1 - \delta.$$

Then we have

$$R_T \leq 4\Delta(\eta) \left( \sqrt{KT \log(\frac{KT}{\delta})} + 4\sqrt{\frac{5}{3}KT \log K} + 8 \log(\frac{KT}{\delta}) \right) \text{ with probability } (1-\delta) \cdot (1-\eta)^T.$$

$\square$

### C.2   PROOF FOR THEOREM 5

**Lemma 2.** *For any non-decreasing differentiable function $\Delta = \Delta(T) > 0$ satisfying*

$$\lim_{T \to \infty} \frac{\Delta(T)^2}{\log(T)} = \infty, \qquad \lim_{T \to \infty} \Delta'(T) \leq C_0 < \infty,$$

*and any $0 < \delta < 1, a > 2$ we have*

$$P\left(R_T \leq \Delta(T) \cdot \log(1/\delta) \cdot O^*(\sqrt{T})\right) \geq (1 - \delta)\left(1 - \frac{1}{T^a}\right)^T$$

*for any $T$ large enough.*

*Proof.* Let $a > 2$ and let us denote

$$F(y) = \int_{-y}^{y} f(x_1, x_2, \ldots, x_K) dx_1 dx_2 \ldots dx_K,$$

$$\zeta(T) = F\left(\Delta(T) \cdot \mathbf{1}\right) - \left(1 - \frac{1}{T^a}\right)$$

for $y \in \mathbb{R}^K$ and $\mathbf{1} = (1, \ldots, 1) \in \mathbb{R}^K$. Let also $y_{-i} = (y_1, \ldots, y_{i-1}, y_{i+1}, \ldots, y_K)$ and $x|_{x_i = y} = (x_1, \ldots, x_{i-1}, y, x_{i+1}, \ldots, x_K)$. We have $\lim_{T \to \infty} \zeta(T) = 0$.

The gradient of $F$ can be estimated as

$$\nabla F \leq \left( \int_{-y_{-1}}^{y_{-1}} f\left(x|_{x_1=y_1}\right) dx_2 \ldots dx_K, \ldots, \int_{-y_{-K}}^{y_{-K}} f\left(x|_{x_K=y_K}\right) dx_1 \ldots dx_{K-1} \right).$$

According to the chain rule and since $\Delta'(T) \geq 0$, we have

$$\frac{dF(\Delta(T) \cdot \mathbf{1})}{dT} \leq \int_{-\Delta(T) \cdot \mathbf{1}_{-1}}^{\Delta(T) \cdot \mathbf{1}_{-1}} f\left(x|_{x_1=\Delta(T)}\right) dx_2 \ldots dx_K \cdot \Delta'(T) +$$

$$\ldots + \int_{-\Delta(T) \cdot \mathbf{1}_{-K}}^{\Delta(T) \cdot \mathbf{1}_{-K}} f\left(x|_{x_K=\Delta(T)}\right) dx_1 \ldots dx_{K-1} \cdot \Delta'(T).$$

Next we consider

$$\int_{-\Delta(T)\mathbf{1}_{-i}}^{\Delta(T)\mathbf{1}_{-i}} f\left(x|_{x_i=\Delta(T)}\right) dx_1 \ldots dx_{i-1} dx_{i+1} \ldots dx_K$$

$$= e^{-\frac{1}{2}a_{ii}(\Delta(T))^2 + \mu_i \Delta(T)} \cdot \int_{-\Delta(T)\mathbf{1}_{-i}}^{\Delta(T)\mathbf{1}_{-i}} e^{g(x_{-i})} dx_1 \ldots dx_{i-1} dx_{i+1} \ldots dx_K.$$

Here $e^{g(x_{-i})}$ is the conditional density function given $x_i = \Delta(T)$ and thus $\int_{-\Delta(T)\mathbf{1}_{-i}}^{\Delta(T)\mathbf{1}_{-i}} e^{g(x_{-i})} dx_1 \ldots dx_{i-1} dx_{i+1} \ldots dx_K \leq 1$. We have

$$\int_{-\Delta(T)\mathbf{1}_{-i}}^{\Delta(T)\mathbf{1}_{-i}} f\left(x|_{x_i=\Delta(T)}\right) dx_1 \ldots dx_{i-1} dx_{i+1} \ldots dx_K$$

$$\leq e^{-\frac{1}{2}a_{ii}(\Delta(T))^2 + \mu_i \Delta(T)}$$

$$\leq e^{-\frac{1}{2}\min_j a_{jj}(\Delta(T))^2 + \max_j \mu_j \Delta(T)}.$$

Then for $T \geq T_0$ we have $\Delta'_T \leq C_0 + 1$ and in turn

$$\zeta'(T) \leq (C_0 + 1) \cdot K \cdot e^{-\frac{1}{2}\min_j a_{jj}(\Delta(T))^2 + \max_j \mu_j \Delta(T)} - a \cdot T^{-a-1}.$$

Since we only consider non-degenerate Gaussian bandits with $\min a_{ii} > 0$, $\mu_i$ are constants and $\Delta(T) \to \infty$ as $T \to \infty$ according to the assumptions in Lemma 2, there exits $C_1 > 0$ and $T_1$ such that

$$e^{-\frac{1}{2}\min_j a_{jj}(\Delta(T))^2 + \max_j \mu_j \Delta(T)} \leq e^{-C_1 \Delta(T)^2} \text{ for every } T > T_1.$$

Since $\lim_{T \to \infty} \frac{\Delta(T)^2}{\log(T)} = \infty$, we have

$$\Delta(T)^2 > \frac{2(a+1)}{C_1} \cdot \log(T) \text{ for } T > T_2.$$

These give us that

$$
\begin{aligned}
\zeta(T)' &\leq (C_0 + 1)Ke^{-2(a+1)\log T} - aT^{-a-1} \\
&= (C_0 + 1)Ke^{-2(a+1)\log T} - ae^{-(a+1)\log T} \\
&< 0 \text{ for } T \geq T_3 \geq \max(T_0, T_1, T_2).
\end{aligned}
$$

This concludes that $\zeta'(T) < 0$ for $T \geq T_3$. We also have $\lim_{T\to\infty} \zeta(T) = 0$ according to the assumptions. Therefore, we finally arrive at $\zeta(T) > 0$ for $T \geq T_3$. This is equivalent to

$$
\int_{-\Delta(T)\cdot\mathbf{1}}^{\Delta(T)\cdot\mathbf{1}} f(x_1, \ldots, x_K)\, dx_1 \ldots dx_K \geq 1 - \frac{1}{T^a},
$$

i.e. the rewards are bounded by $\Delta(T)$ with probability $1 - \frac{1}{T^a}$. Then by the same argument for $T$ large enough as in the proof of Theorem 4, we have

$$
P\left(R_T \leq \Delta(T) \cdot \log(1/\delta) \cdot O^*(\sqrt{T})\right) \geq (1 - \delta)(1 - \frac{1}{T^a})^T.
$$

$\square$

*Proof of Theorem 5.* In Lemma 2, we choose $\Delta(T) = \log(T)$, which meets all of the assumptions. The result now follows from $\log T \cdot O^*(\sqrt{T}) = O^*(\sqrt{T})$, Lemma 2 and Theorem 4. $\square$

### C.3 PROOF FOR THEOREM 6

We first list 3 known lemmas. The following lemma by Duchi (2009) provides a way to bound deviations.

**Lemma 3.** *For any function class F, and i.i.d. random variable $\{x_1, x_2, \ldots, x_T\}$, the result*

$$
E_x\left[\sup_{f\in F}\left|E_x f - \frac{1}{T}\sum_{t=1}^T f(x_t)\right|\right] \leq 2R_T^c(F)
$$

*holds where $R_T^c(F) = E_{x,\sigma}\left[\sup_f\left|\frac{1}{T}\sum_{t=1}^T \sigma_t f(x_t)\right|\right]$ and $\sigma_t$ is a $\{-1, 1\}$ random walk of t steps.*

The following result holds according to Balcan (2011).

**Lemma 4.** *For any subclass $A \subset F$, we have $\hat{R}_T^c \leq R(A, T) \cdot \frac{\sqrt{2\log|A|}}{T}$, where $R(A, T) = \sup_{f\in A}\left(\sum_{t=1}^T f(x_t)\right)^{\frac{1}{2}}$ and $\hat{R}_T^c = \sup_f\left|\frac{1}{T}\sum_{t=1}^T \sigma_t f(x_t)\right|$.*

A random variable $X$ is $\sigma^2$-sub-Gaussian if for any $t > 0$, the tail probability satisfies

$$
P(|X| > t) \leq Be^{-\sigma^2 t^2},
$$

where $B$ is a positive constant. The following lemma is listed in the Appendix A of Chatterjee (2014).

**Lemma 5.** *For i.i.d. $\sigma^2$-sub-Gaussian random variables $\{Y_1, Y_2, \ldots, Y_T\}$, we have*

$$
E\left[\max_{1\leq t\leq T}|Y_t|\right] \leq \sigma\sqrt{2\log T} + \frac{4\sigma}{\sqrt{2\log T}}.
$$

*Proof for Theorem 6.* Let us define $F = \{f_j : x \to x_j | j = 1, 2, \ldots, K\}$. Let $x_t = (r_1^t, r_2^t, \ldots, r_K^t)$ where $r_i^t$ is the reward of arm $i$ at step $t$ and let $a_t$ be the arm selected at time $t$ by EXP3.P. Then for any $f_j \in F$, $f_j(x_t) = r_j^t$. In Gaussian-MAB, $\{x_1, x_2, \ldots, x_T\}$ are i.i.d. random variables since the Gaussian distribution $\mathcal{N}(\mu, \Sigma)$ is invariant to time and independent of time. Then by Lemma 3, we have

$$
E\left[\max_i\left|\mu_i - \frac{1}{T}\sum_{t=1}^T r_i^t\right|\right] \leq 2R_T^c(F).
$$

We consider

$$
\begin{aligned}
E\left[|R'_T - R_T|\right] &= E\left[\left|T \cdot \max_i \mu_i - \sum_{t=1}^{T} \mu_{a_t} - \left(\max_i \sum_{t=1}^{T} r_i^t - \sum_{t=1}^{T} r_{a_t}^t\right)\right|\right] \\
&= E\left[\left|T \cdot \max_i \mu_i - \max_i \sum_{t=1}^{T} r_i^t - \left(\sum_{t=1}^{T} \mu_{a_t} - \sum_{t=1}^{T} r_{a_t}^t\right)\right|\right] \\
&\leq E\left[\left|T \cdot \max_i \mu_i - \max_i \sum_{t=1}^{T} r_i^t\right|\right] + E\left[\left|\sum_{t=1}^{T} \mu_{a_t} - \sum_{t=1}^{T} r_{a_t}^t\right|\right] \\
&\leq E\left[\max_i \left|T \cdot \mu_i - \sum_{t=1}^{T} r_i^t\right|\right] + E\left[\left|\sum_{t=1}^{T} \mu_{a_t} - \sum_{t=1}^{T} r_{a_t}^t\right|\right] \\
&\leq 2T R_T^c(F) + 2T_1 R_{T_1}^c(F) + \cdots + 2T_K R_{T_K}^c(F)
\end{aligned}
\tag{3}
$$

where $T_i$ is the number of pulls of arm $i$. Clearly $T_1 + T_2 + \ldots + T_K = T$. By Lemma 4 with $A = F$ we get

$$
\begin{aligned}
R_T^c(F) &= E\left[\hat{R}_T^c(F)\right] \leq E[R(F,T)] \cdot \frac{\sqrt{2 \log K}}{T}, \\
R_{T_i}^c(F) &\leq E\left[R\left(F, T_i\right)\right] \cdot \frac{\sqrt{2 \log K}}{T_i} \qquad i = \{1, 2, ,\ldots, K\}.
\end{aligned}
$$

Since $R(F,T)$ is increasing in $T$ and $T_i \leq T$, we have $R_{T_i}^c(F) \leq E\left[R\left(F,T\right)\right] \cdot \frac{\sqrt{2 \log K}}{T_i}$.

We next bound the expected deviation $E\left[|R'_T - R_T|\right]$ based on (3) as follows

$$
\begin{aligned}
E\left[|R'_T - R_T|\right] &\leq 2T E[R(F,T)] \frac{\sqrt{2 \log K}}{T} + \sum_{i=1}^{K}\left[2T_i E[R(F,T)] \frac{\sqrt{2 \log K}}{T_i}\right] \\
&\leq 2(K+1)\sqrt{2 \log K} E[R(F,T)].
\end{aligned}
\tag{4}
$$

Regarding $E[R(F,T)]$, we have

$$
\begin{aligned}
E[R(F,T)] &= E\left[\sup_{f \in F}\left(\sum_{t=1}^{T} f(x_t)\right)^{\frac{1}{2}}\right] = E\left[\sup_i\left(\sum_{t=1}^{T}(r_i^t)^2\right)^{\frac{1}{2}}\right] \\
&\leq E\left[\sum_{i=1}^{K}\left(\sum_{t=1}^{T}(r_i^t)^2\right)^{\frac{1}{2}}\right] \leq \sum_{i=1}^{K} E\left[\left(T \cdot \max_{1 \leq t \leq T}(r_t^i)^2\right)^{\frac{1}{2}}\right] \\
&= \sqrt{T} \cdot \sum_{i=1}^{K} E\left[\max_{1 \leq t \leq T}|r_i^t|\right].
\end{aligned}
\tag{5}
$$

We next use Lemma 5 for any arm $i$. To this end let $Y_t = r_i^t$. Since $x_t$ are Gaussian, the marginals $Y_t$ are also Gaussian with mean $\mu_i$ and standard deviation of $a_{ii}$. Combining this with the fact that a Gaussian random variable is also $\sigma^2$-sub-Gaussian justifies the use of the lemma. Thus $E\left[\max_{1 \leq j \leq T}|r_i^j|\right] \leq a_{i,i} \cdot \sqrt{2 \log T} + \frac{4a_{i,i}}{\sqrt{2 \log T}}$.

Continuing with equation 5 we further obtain

$$
\begin{aligned}
E[R(F,T)] &\leq \sqrt{T} \cdot K \cdot \max_i\left(a_{i,i}\sqrt{2 \log T} + \frac{4a_{i,i}}{\sqrt{2 \log T}}\right) \\
&= \left(K\sqrt{2T \log T} + \frac{4\sqrt{T}}{\sqrt{2 \log T}}\right) \cdot \max_i a_{i,i}.
\end{aligned}
\tag{6}
$$

By combining equation 4 and equation 6 we conclude

$$E\left[|R_T' - R_T|\right] \le 2(K+1)\sqrt{2\log K} \cdot \max_i a_{i,i} \cdot \left(K\sqrt{2T\log T} + \frac{4\sqrt{T}}{\sqrt{2\log T}}\right) \tag{7}$$

$$= O^*(\sqrt{T}).$$

We now turn our attention to the expectation of regret $E[R_T]$. It can be written as

$$E\left[R_T\right] = E\left[R_T \mathbb{1}_{R_T \le O^*(\sqrt{T})}\right] + E\left[R_T \mathbb{1}_{R_T > O^*(\sqrt{T})}\right]$$

$$\le O^*(\sqrt{T})P\left(R_T \le O^*(\sqrt{T})\right) + E\left[R_T \mathbb{1}_{R_T > O^*(\sqrt{T})}\right] \le O^*(\sqrt{T}) + E\left[R_T \mathbb{1}_{R_T > O^*(\sqrt{T})}\right]$$

$$= O^*(\sqrt{T}) + E\left[R_T \mathbb{1}_{O^*(\sqrt{T}) < R_T < O^*(\sqrt{T}) + E[R_T]}\right] + E\left[R_T \mathbb{1}_{R_T \ge O^*(\sqrt{T}) + E[R_T]}\right]. \tag{8}$$

We consider $\delta = 1/\sqrt{T}$ and $\eta = T^{-a}$ for $a > 2$. We have

$$\lim_{T\to\infty} (1-\delta)(1-\eta)^T = \lim_{T\to\infty} (1-\delta)(1 - \frac{1}{T^a})^T$$

$$= \lim_{T\to\infty} (1-\delta)(1 - \frac{1}{T^a})^{(T^a) \cdot \frac{T}{T^a}} = \lim_{T\to\infty} e^{\frac{T}{T^a}}$$

and

$$\lim_{T\to\infty} \left(1 - (1-\delta)(1-\eta)^T\right) \cdot \log T \cdot T = \lim_{T\to\infty} (1 - e^{\frac{T}{T^a}}) \cdot \log(T) \cdot T$$

$$\le \lim_{T\to\infty} \log(T) \cdot T \cdot T^{1-a} = \lim_{T\to\infty} T^{2-a} \cdot \log(T) = 0. \tag{9}$$

Let $P_1 = P\left(R_T \le \log(1/\delta)O^*(\sqrt{T})\right)$ which equals to $P\left(R_T \le O^*(\sqrt{T})\right)$ since $\log(1/\delta) = \log(\sqrt{T}) = O^*(\sqrt{T})$. By Theorem 5 we have $P_1 = (1-\delta) \cdot (1-\eta)^T$.

Note that $E[R_T] \le C_0 \log(T) \cdot T$ as shown by

$$E[R_T] = E\left[\max_i \sum_{t=1}^T r_i^t - \sum_{t=1}^T r_{a_t}^t\right] \le 2E\left[\max_i \sum_{t=1}^T |r_i^t|\right] \le 2T \cdot E\left[\max_i \max_t |r_i^t|\right]$$

$$\le 2T \cdot \sum_{i=1}^K E\left[\max_t |r_i^t|\right] \le 2T \cdot \sum_{i=1}^K \left(a_{i,i}\sqrt{2\log T} + \frac{4a_{i,i}}{\sqrt{\log T}}\right)$$

$$\le 2T \cdot \sum_{i=1}^K \max_i a_{i,i} \left(\sqrt{2\log T} + \frac{4}{\sqrt{\log T}}\right)$$

$$\le C_0 \cdot T \cdot \log(T)$$

for a constant $C_0$.

The asymptotic behavior of the second term in equation 8 reads

$$E\left[R_T \mathbb{1}_{O^*(\sqrt{T}) < R_T < O^*(\sqrt{T}) + E[R_T]}\right] = E\left[R_T \mathbb{1}_{R_T - O^*(\sqrt{T}) \in (0, E[R_T])}\right]$$

$$= E\left[\left(R_T - O^*(\sqrt{T})\right) \mathbb{1}_{R_T - O^*(\sqrt{T}) \in (0, E[R_T])}\right] + O^*(\sqrt{T})$$

$$\le E[R_T] P\left(R_T - O^*(\sqrt{T}) \in (0, E[R_T])\right) + O^*(\sqrt{T})$$

$$\le E[R_T] P\left(R_T - O^*(\sqrt{T}) > 0\right) + O^*(\sqrt{T})$$

$$\le C_0 \log(T) \cdot T \cdot (1 - P_1) + O^*(\sqrt{T}) = O^*(\sqrt{T})$$

where at the end we use equation 9.

Regarding the third term in equation 8, we note that $R'_T \leq E[R_T]$ by the Jensen's inequality. By using equation 7 and again equation 9 we obtain

$$
\begin{aligned}
& E\left[R_T \mathbb{1}_{R_T \geq O^*(\sqrt{T}) + E[R_T]}\right] \\
&= E\left[(R_T - R'_T)\, \mathbb{1}_{(R_T - E[R_T]) \geq O^*(\sqrt{T})}\right] + E\left[R'_T \mathbb{1}_{(R_T - E[R_T]) \geq O^*(\sqrt{T})}\right] \\
&\leq E\left[\|R_T - R'_T\|\right] + R'_T \cdot P\left(R_T \geq E[R_T] + O^*(\sqrt{T})\right) \\
&\leq E\left[\|R_T - R'_T\|\right] + E[R_T] \cdot P\left(R_T \geq E[R_T] + O^*(\sqrt{T})\right) \\
&\leq O^*(\sqrt{T}) + C_0 \cdot \log(T) \cdot T \cdot P\left(R_T \geq O^*(\sqrt{T})\right) \\
&= O^*(\sqrt{T}) + C_0 \cdot \log(T) \cdot T\,(1 - P_1) = O^*(\sqrt{T}).
\end{aligned}
$$

Combining all these together we obtain $E[R_T] = O^*(\sqrt{T})$ which concludes the proof. $\qquad\square$