# OpenReview forum: "Regret Bounds and Reinforcement Learning Exploration of EXP-based Algorithms"
_ICLR.cc/2021/Conference — Reject_

### Official Review · AnonReviewer4 · 2020-10-13
**This paper develops EXP4-style algorithms for Gaussian bandit and RL, proving upper and lower bounds and give empirical evaluations. I have a few questions:**

**Rating:** 4
**Confidence:** 3

**Review:**

This paper develops EXP4-style algorithms for Gaussian bandit and RL, proving upper and lower bounds and give empirical evaluations. I have a few questions:

1. I think MAB with unbounded loss is a solved problem? At least these regret bound should be known. For example , see [Hannan consistency in on-line learning in case of unbounded losses under partial monitoring]
2. The RL part is even more confusing. Say we have n experts:
(1) when n is big, then we need to maintain n DQN simultaneously, which sounds unrealistic.
(2) when n is small, as in this paper, a good ensemble can only guarantee you get to match the better one of the two experts.  However, a simple way to do that is to train them and evaluate the policies separately. A lot of technicalities can be avoided this way. So why bother using EXP4?
3. Actually many existing algorithms, at least in tabular RL, are inspired by EXP4. Indeed, consider the policy gradient method on a MAB, then you get an EXP-style algorithm (of course this also depends on the feedback model). See [On the Theory of Policy Gradient Methods: Optimality, Approximation, and Distribution Shift] for example. More sophisticated ways like the Nash V-learning algorithm in [ Near-Optimal Reinforcement Learning with Self-Play], could be applied to solve competitive RL problems.

Of course, these papers are for tabular RL, which is too simplified. How to use function approximation with EXP4 is an interesting yet challenging question. I think the main weakness of the approaches in this paper is that, it is not clear why exploration with EXP4 is more desirable and why a naive combination with DQN should work. Indeed, I think this comination can be very data inefficient.

---

> ### Author Response · Authors · 2020-11-22
> **Perspectives on the upper bound, motivation and experiment of the paper**
>
> Dear reviewer,
>
> Thank you very much for providing these very detailed comments. We highly value your feedback and appreciate your time and effort. Below are responses and our perspectives regarding the comments.
>
> Regarding the unbounded rewards, reward in our case can be arbitrarily large for every time step and we derive the asymptotic $O^*(\sqrt T)$ upper bound on regret. We acknowledge providing the reference paper. The result in the paper relies on the assumption that  $r_t \leq t^\alpha$ where $r_t$ is the reward at time step $t$ and $a$ is a constant. This adds a step-by-step upper bound on rewards. For small $t$, the upper bound on reward can be small. So the underlying assumption of the paper is much more stringent than our assumption of arbitrary unbounded reward.
>
> For the experts in EXP4-RL, the ultimate purpose of the paper is to achieve a better exploration and thereby a higher cumulative reward by combining different experts in an innovative way. The numerical performance of EXP4-RL is better than the performance of RND, the best previously known algorithm. EXP4-RL works best if we already have a SOTA algorithm (RND in this paper) as an expert; we can improve on the SOTA and upgrade the SOTA. Therefore, the number of experts $n$ is not required to be very large, as long as a SOTA is included. With a small $n$, the efficiency of EXP4-RL does not decrease.
>
> Thanks very much for providing these great sources of EXP4-based exploration. They are for tableau RL which have very limited practical relevance. Our case of non-tableau RL is still a novel approach.
>
> For the reason why EXP4 works, we hope that the following perspective is enlightening. The original EXP4 algorithm combines experts and is comparable to the best expert theoretically with a sublinear regret. There are no restrictions on these experts. Then it’s natural to use $Q$-neural networks as experts when adapting EXP4 to RL. We expect to have a performance comparable and even better than the best expert (SOTA). The numerical performance verifies this intuition and shows that EXP4-RL outperforms the best expert. Let us consider that there is uniform sampling among experts in the original EXP4 case. Then it’s not guaranteed to be comparable to the best expert (SOTA) if it does not include a suboptimal expert, since the overall performance would be degraded by the worst expert with a constant proportion. Actually, bootstrapped DQN (Osband et.al. 2016) utilizes uniform sampling in DQNs and even adds a carefully chosen mask function for those DQNs. But its performance on Montezuma’s Revenge is inferior to RND. These facts explain why uniform sampling of $Q$-networks does not work.
>
> We highly value your suggestions and we hope we provide adequate responses and arguments.

---

### Official Review · AnonReviewer2 · 2020-10-29
**This paper studied EXP algorithms from interesting angles but its results are not strong enough**

**Rating:** 4
**Confidence:** 4

**Review:**

This paper contributes to the study of EXP-based algorithms in two aspects. One is on the theoretical aspect: It analyzes the lower and upper bounds of EXP-3 for Gaussian multi-bandit setting for which the reward can be unbounded. The other is on the empirical aspect: It applied EXP4, originally developed for MAB, to Rl applications and demonstrated its advanced empirical performance.

This paper is overall clearly written. However, my general feeling is EXP3 part and EXP4 part are quite separated and have different flavors. Other than both are algorithms in the EXP family, I did not find much connection between them.


For the EXP3P part, I have difficulty to appreciate the value of a linear-in-T lower bound for small T. (And this small T range further depends on bandit setting such as \mu, q). In fact, the lower bound has in Theorem also involves \epsilon, which together with T are restricted by an inequality. If I further manipulate the lower bound expression and the required T range, I think it is not hard to obtain quadric or cubic in T lower bound. (One extremal but obviously non-interesting lower bound is  constant*T^n for any  with T<=1).  So I really want to understand why a bound for small T is useful? Even if it is useful, why the order regarding T matters?

The lower bound part also has some basic conceptual mistakes.  When you study a lower bound, please use \Omega() or \Theta() instead of O(). You can say your regret has a lower bound given by \Theta() or simply say your regret = (or \in) \Omega().  But don't say your regret has a lower bound O(), which means nothing. (Technically, T^2 has a lower bound O(T^3) is still correct because T = (or \in) O(T^3).   Further, big O notation is asymptotical notation, i.e., meaningful only for large T. I don't think you can use O(T) to represent a lower bound for small T.


The experiments only compare Alg 2 with RND. What if you consider a baseline that involve multiple different DQNs and randomizes/explores among them?

---

> ### Author Response · Authors · 2020-11-22
> **Perspectives on the structure, lower bound and baseline of the paper**
>
> Dear reviewer,
>
> Thank you very much for providing these very detailed comments. We highly value your feedback and appreciate your time and effort. Below are responses and our perspectives regarding the comments.
>
> Regarding the connection between EXP3 and EXP4, the former is typically used and analyzed in the MAB setting, while the latter is studied in the contextual bandit setting. We analyze EXP3 but use EXP4 in the algorithm since RL employs states which correspond to context in EXP4. All previous results for EXP3 assume bounded reward to achieve regret guarantee. RL has unbounded reward and states, which makes it suitable for EXP4 with unbounded reward. Since analyzing algorithms depending on states is very hard, we analytically study EXP’s related algorithm EXP3 in the unbounded reward setting with no notion of states (MAB). We show that EXP3.P in the MAB case is optimal even with unbounded rewards.
>
> Regarding the lower bound result in the paper, we establish a lower bound on regret by means of Gaussian instances, which is different from existing results assuming bounded rewards. The state-of-art result relies on the $[0,1]$ reward assumption. Our lower bound holds for unbounded rewards, though at the cost of the bounded time horizon. This time horizon can be chosen to be very large by controlling the constant in the linear factor.
>
> Regarding the baseline, RND is known to produce optimal solutions for exploration on a lot of Atari games; we show that our EXP4-RL has better performance than RND. This should be the case since EXP4 in MAB achieves optimal performance (it matches the performance of the optimal expert with respect to regret). If we consider an ensemble of DQNs and do uniform sampling, we cannot expect it to be performing well. The overall performance would be degraded by the worst expert due to uniform sampling and cannot be comparable to the best expert. Actually, bootstrapped DQN (Osband et.al. 2016) utilizes uniform sampling in DQNs and even adds a carefully chosen mask function for those DQNs. But its performance on Montezuma’s Revenge is inferior to RND. We hope that this explanation helps interpreting the intuition of EXP4, why EXP4 works rather than uniform sampling, and why we use RND as a baseline instead of uniform sampling of DQNs.
>
> Regarding the format of the lower bound, we apologize for some sloppy notation which has already been revised in the paper. Thanks for pointing the mistakes out. We have also fixed the typos.
>
> We sincerely appreciate your effort and time.

---

### Official Review · AnonReviewer1 · 2020-10-30
**Contributions appear marginal, and some doubts about the regret lower bound.**

**Rating:** 4
**Confidence:** 3

**Review:**

The authors consider analyzing the EXP3.P algorithm for the case of unbounded reward functions, in the sense that the rewards are governed by a Gaussian distribution. The authors first demonstrate a regret lower bound result on the Gaussian MABs when the time horizon is bounded from above. Then, the authors proceed to the analysis of the EXP3.P algorithm on the Gaussian MABs, and establish a regret bound similar to that of Auer et al. 2002. Finally, the authors apply the EXP3.P, where an expert corresponds to a Q-learning network, in the EXP4-RL algorithm, and evaluate it on multiple RL instances.

Major comments: The major technical contributions seem to be the regret bound for EXP3.P for the case of Gaussian MAB, as stated in Theorem 4. Based on the authors' notation in the first paragraph of page 3, the Gaussian reward distribution is stationary across time. This contribution appears marginal, since the Theorem appears to be a straightforward consequence of the EXP3.P regret by Auer et al. 2002 by conditioning on all realized rewards to lie in [-\Delta, \Delta]. The technical part on how to identify the best expert is already dealt with by the analysis in Auer et al. 2002.

Another contributions are Theorems 1-3, which are regret lower bounds for Gaussian MABs. I am not sure how to interpret these regret lower bounds, since they require the horizon length to be bounded from above. More precisely, the authors show that for any algorithm, there exists a Gaussian MAB instance such that $\text{Reg}(T) \geq c T$ when $T\leq C$, where $c, C$ are instance-dependent  constants. While this bound is a mathematically sound statement, it does not imply anything about the difficulty of the underlying problem when T is large. For example, for regret upper bound of an MAB algorithm, one almost always establishes a guarantee of the form $\text{Reg}(T) \leq \text{Bound}(T)$ for all $T \geq C'$, where $C'$ an instance dependent constant. I am not too sure what is the message the authors are trying to convey here, since we know that the state-of-the-art regret lower bound is $\Omega(\sqrt{KT})$ for sufficiently large T.

Finally, if I understand the underlying motivation of the authors correctly, the ultimate problem that the authors are trying to address seems to be a stochastic best arm identification problem with (sub-)Gaussian rewards, where an arm here corresponds to a Q-network. I am not sure why the authors resort to EXP type algorithms for a stochastic problem.

Minor Comments:

I believe that the inequality w RL(T) ≥ O(√T · γ ) on page 4 should be \leq.

In Algorithm 1, in the initialization, w_i should be replaced by w_i(1).

In Algorithm 2, it requires to compute  y_k = E[ˆx_{kj} ], and the authors should elaborate on how the expectation is computed.

In general, there are quite a few typos, and some parts of the writing are  a bit ambiguous in the way they are phrased. I advise the authors to proofread and also polish the writing.

---

> ### Author Response · Authors · 2020-11-22
> **Perspectives on lower bound, upper bound and the motivation of the paper**
>
> Dear reviewer,
>
> Thank you very much for providing these very detailed comments. We highly value your feedback and appreciate your time and effort. Below are responses and our perspectives regarding the comments.
>
> For the upper bound result in the paper, first of all, it doesn’t rely on the bounded assumption. By truncating the reward with $[-\Delta,\Delta]$, we get that the regret can be bounded above by $\Delta \cdot \Omega(\sqrt T )$ where $\Omega(\sqrt T)$ is derived in (Auer et.al 2002), with a certain probability. However, this in the regret term is rather large to guarantee that such probability is close to 1. To deal with the trade-off, we choose $\Delta$ to be $\log T$, while maintaining an asymptotically probability 1. This part needs substantial additional work. Moreover, since regret can be unbounded due to the unbounded reward, this high-probability regret bound provides no information on the expected value of the regret, which is very important for quantifying the overall regret of the algorithm and is the most technical part. We utilize Randemacher complexity in a novel way to interpret the regret and establish an asymptotic $O^*(\sqrt T)$ result, which shares the same order as the high-probability regret. This provides the proof that even if the reward is unbounded, the high-probability regret and the expected value of the regret are still bounded above by $O^*(\sqrt T)$ asymptotically, coinciding with the state-of-art result in the bounded reward case. We hope that this explanation demonstrates the contributions of the theoretical part related to the upper bound of the regret.
>
> For the lower bound result in the paper, we show a lower bound $c \cdot T$ for $T \leq C$ for MAB with unbounded rewards. However, this constant $C$ can be pretty large as long as the parameter $c$ is small. So by properly choosing  $c$ and $C$, the lower bound $c \cdot T$ can hold for large $T$. From this perspective, this lower bound implies a regret order for large $T$. For the known result $\Omega(\sqrt T)$ for any T, it requires the reward to be bounded, which is not the case in our work.
>
> The ultimate purpose of this work is to apply EXP4 from MAB to RL that potentially has unbounded rewards. Since previously studied EXP-series algorithms consider the [0,1] reward assumption, we are able to provide its theoretical guarantee in MAB with unbounded rewards. In summary, the theoretical study of EXP for MAB in the unbounded case links to the EXP4-RL algorithm where reward is unbounded.
>
> Thank you very much for pointing out the typos; we have fixed them in the revised paper.Regarding the expression of the expected value, $y_k=E[\hat{x}_{kj}  \cdot P^k_j]$ where $P^k_j$ is the probability of choosing action $j$ given by expert, is computed by applying $\epsilon$-greedy to $Q_k$-network.
>
> Overall, we demonstrate the motivation and contributions of the paper from the above perspectives.

---

### Decision · Program_Chairs · 2021-01-07
**Final Decision**

**Decision:**

Reject

**Comment:**

This paper consider a classical multi-armed bandit problem (then a more general RL setting) and prove some upper and lower bounds, in cases that were not explicitly studied in the literature.

However, those results are very incremental and do not justify (maybe yet, going beyond the sub-Gaussian case could be interesting) yet acceptance.